# Association between participation in the Northern Finland Birth Cohort 1966 study and use of psychiatric care services

**Martta Kerkelä**[1]*, **Mika Gissler**[2,3,4,5], **Tanja Nordström**[6,7,8], **Juha Veijola**[1,8,9]

**1** Research Unit of Clinical Neuroscience, University of Oulu, Oulu, Finland, **2** Department of Knowledge Brokers, Finnish Institute for Health and Welfare, Helsinki, Finland, **3** Research Centre for Child Psychiatry, University of Turku, Turku, Finland, **4** Academic Primary Health Care Centre, Region Stockholm, Stockholm, Sweden, **5** Department of Molecular Medicine and Surgery, Karolinska Institute, Stockholm, Sweden, **6** Northern Finland Birth Cohorts, Arctic Biobank, Infrastructure for Population Studies, University of Oulu, Oulu, Finland, **7** Research Unit of Population Health, University of Oulu, Oulu, Finland, **8** Medical Research Center, University Hospital and University of Oulu, Oulu, Finland, **9** Department of Psychiatry, University Hospital of Oulu, Oulu, Finland

* martta.kerkela@oulu.fi

**Data Availability Statement:** Data cannot be shared publicly because the utilised data have been given for this specific study (Permission numbers: TK-53-1053-19, VRK/3285/2019-2, THL/1254/

## Abstract

### Aims

In most population-based epidemiological follow-up studies the aim is not to intervene in the life of the participants. Although the idea is not to intervene, being a member of the longitudinal follow-up study and studies conducted during follow-up may affect the target population. A population-based study including mental health enquiries might reduce the unmet need for psychiatric treatment by motivating people to seek treatment for their psychiatric ill-health. We examined the use of psychiatric care services in the population born in the year 1966 in Northern Finland, of whom 96.3% are participants in the prospective Northern Finland Birth Cohort 1966 (NFBC1966).

### Methods

As a study cohort we used people born in 1966 in Northern Finland ($n = 11\,447$). The comparison cohort included all the people born in the years 1965 and 1967 in the same geographical area ($n = 23\,339$). The follow-up period was from age 10 to 50 years. The outcome measure was the use of psychiatric care services, which was analysed using Cox Proportional Hazard regression and Zero-Truncated Negative Binomial Regression.

### Results

People born in 1966 in Northern Finland did not differ from those born in 1965 and 1967 in terms of the outcome measure.

### Conclusions

We found no association between participation in an epidemiological follow-up study and the use of psychiatric care services. The NFBC1966 may be regarded as a representative

2019), and the data cannot be shared without permission from the register keepers. The same data can be applied from Findata, the Finnish Health and Social Data Permit Authority by researchers who meet the criteria for access to confidential data. Additionally, Findata, can prepare statistical tables from the registers used in this study: https://findata.fi/en/permits/data-requests/.

**Funding:** The author(s) received no specific funding for this work.

**Competing interests:** The authors have declared that no competing interests exist.

at the population level in terms of psychiatric outcomes despite the personal follow-up of the birth cohort. The associations of participation in epidemiological follow-up studies have previously been under-examined, and the results need to be replicated.

## Introduction

Population-based follow-up studies are epidemiological studies in which a defined population is selected and followed up for longitudinal assessment of exposure-outcome relations [1]. In most population-based follow-up studies the aim is not to intervene in the life of the participants, so that the sample is a representative sample of the population. Although the idea is not to intervene, being a member of the longitudinal follow-up study and studies conducted during follow-up may affect the study population in various ways.

The awareness of participation in the study can be described as the Hawthorne effect, which describes the awareness of the study participants being studied and the possible impact on their behaviour [2]. In the Derbyshire Smoking Study, the Hawthorne effect of repeatedly measured smoking behaviour of about 6 000 adolescents was studied by questionnaire yearly from 1974 to 1978 in selected schools. The self-reported smoking habits were compared to adolescents whose smoking habits were measured first time in 1978. The findings revealed that the prevalence of smoking was lower in those schools, which had been surveyed for five years. [3]. Berkhout et al. pooled the findings of 15 studies published in 2012–2022 on the Hawthorne effect in primary care or outpatient clinics or healthy subjects. In the meta-analysis, the Hawthorne effect was defined as "an aware or unconscious complex behavioural change in a study environment, related to the interaction of four biases affecting the study subjects and investigators: selection bias, commitment and congruence bias, conformity and social desirability bias, and observation and measurement bias". The findings demonstrate that medical research is unavoidably prone to the Hawthorne effect, which restricts its external validity, beginning with the intentional or unintentional selection of the study population, resulting in blind spots in medical knowledge [4].

Epidemiological studies have shown that there is an unmet need for psychiatric treatment in the population [5]. Various proportions of the population have been detected to have psychiatric conditions or disorders and have not been treated at all [6]. A population study including mental health enquiries might reduce the unmet need for psychiatric treatment by motivating people to seek treatment for their psychiatric ill-health. In a population-based Finnish study (the UKKI-study), the stability of and changes in mental health in the adult Finnish population was explored during 1970–1987. A population sample of the UKKI study was followed for 16 years using questionnaires, interviews, and data collection from health care registers. The possible effect of the study was examined by using controls matched to the study population. The UKKI-study participants had used mental health care services more often than the control group during the follow-up. This result may be regarded, at least partly, as an effect of the study itself [7,8].

Reviews of mental health promotion strategies, programmes and interventions have emphasised their importance for mental health, with growing evidence on their effectiveness [9]. Mental health promotion interventions are applicable to individuals and whole populations [10,11], and can be delivered in various ways, including communication technology, home-based interventions, school/workplace interventions and community-based interventions [10]. Low-level digital mental health interventions, without face-to-face interaction, are

proven to improve depression, anxiety and psychological well-being among college students [12], and mental health interventions delivered via smartphone to reduce depressive symptoms in adulthood [13]. The Rural Mental Health Project in Ireland aimed to promote positive mental health in disadvantaged rural communities through information and general awareness-raising activities, community education workshops and structured positive mental health promotion programmes. The findings suggest that the project has had positive effects on community awareness and attitudes at the wider community level, including increased awareness regarding depression and suicide and improved attitudes to help-seeking [14].

We had a unique chance to examine the use of psychiatric care in the population born in 1966, of whom almost all (96.3%) participated originally in the large prospective Northern Finland Birth Cohort 1966 (NFBC1966) [15]. We examined the effects of being a member of the longitudinal follow-up study cohort compared to a population born in the same geographical area in Northern Finland in 1965 and 1967.

## Materials and methods

### Follow-ups of the NFBC1966

The NFBC1966 covers people whose expected date of birth was in 1966 in the former two northernmost provinces in Finland, Oulu and Lapland. The cohort included 12 055 mothers and 12 231 children. Of the NFBC1966 members, 171 (1.5%) were born in 1965, 11 999 (97.9%) in 1966 and 61 (0.5%) in 1967. NFBC1966 is a longitudinal and prospective birth cohort with several follow-ups [16]. The follow-ups were regarded as interventions in the present study.

The first follow-up for the NFBC1966 cohort was conducted during and just after the mother's pregnancy. The follow-up included a questionnaire (including questions on background, life situation and living habits) during pregnancy (from the 24th to the 28th gestational week) collected by midwives in antenatal clinics. The questionnaire included items on depressed mood and smoking during pregnancy. Information on delivery was also collected [17]. The second follow-up was conducted at age 1 year, including a questionnaire concerning children's growth, health, and development from children's welfare clinics, with a 91.2% participation rate [18]. The next follow-up was at age 14 years, with a participation rate of 93.6%. The follow-up was conducted through a postal questionnaire, including questions for the NFBC 1966 members of their growth and health, living habits (questions on smoking, alcohol, and other intoxicant use), school performance and family situation [19].

The next follow-up for the whole cohort was conducted at age 31. The follow-up was done through a postal questionnaire (participation rate 77.4%), including questions on life situation, background information, exercise and physical performance, occupation, living environment, health, use of depression, tranquilliser and sleep medication, questions on possible psychiatric disorder, and living habits (questions on smoking, alcohol and other intoxicant use), and clinical examination (participation rate 71.3%) [20]. The questionnaire included The Hopkins Symptom Checklist-25 (HSCL-25) [21]. During the clinical examination, an additional questionnaire was given to the participants. The additional questionnaire included the Bipolar II scale (BIP2), Hypomanic Personality Scale (HPS), Physical Anhedonia Scale (PAS), Social Anhedonia Scale (SAS), Perceptual Aberration Scale (PER) and Schizoid Scale (SCHD) [22], Tridimensional Personality Questionnaire (TPQ) and Temperament and Character Inventory (TCI) [23].

The latest follow-up was conducted at age 46, with a 66.5% participation rate in the questionnaire and 56.7% in the clinical examination (including the cognitive Paired Associative Learning (PAL) test [24]). The questionnaire included questions on background, lifestyle

**Table 1. Follow-ups of the Northern Finland Birth Cohort 1966 (NFBC 1966).**

| Age (main reference) | Target Population (N) | Questionnaire data (Participation rate) | Psychiatric items in the Questionnaires | Clinical Examination (Participation rate) |
|---|---|---|---|---|
| Antenatal period [17] | All births in the area in 1966 $n = 12\,527$ | Questionnaire to mothers from local midwives during 24th to 28th gestational week $n = 12\,231$ (96.3%) | Question of depression during pregnancy | Information on delivery |
| 1 year [18] | Children alive $n = 11\,870$ | On the children's health and development $n = 10\,821$ (91.2%) | N/A | Clinical examination $n = 10\,821$ (91.2%) |
| 14 years [19] | Children alive $n = 11\,778$ | On growth and health, hobbies, school and family situation $n = 11\,010$ (93.6%) | Questions on smoking, alcohol and other intoxicant use | N/A |
| 31 years [20] | Subjects alive $n = 11\,637$ | On background information, physical health, occupation, health and living habits $n = 8\,767$ (75.3%) | Questions on smoking, alcohol and other intoxicant use, use of mental health services and self-reported mental health problems. HSCL-25-scale. Additional questionnaire including screenings: BIP2, HPS, PAS, SAS, PER, SCHD, TPQ, TCI | Clinical examination $n = 6\,033$ (71.3%) |
| 46 years [26] | Alive and address known in Finland $n = 10\,321$ | On background information, lifestyle, economy, mental resources and health $n = 6\,868$ (66.5%) | Questions on smoking, alcohol and other intoxicant use. HSCL-25, BDI-21, GAD-7, FABQ scales, PAL-test | Clinical examination $n = 5\,861$ (56.7%), Dental clinical examination n = 1964 |

HSCL-25 = Hopkins Symptom Checklist-25, BIP2 = Bipolar II scale, HPS = Hypomanic Personality Scale, PAS = Physical Anhedonia Scale, SAS = Social Anhedonia Scale, PER = Perceptual Aberration Scale, SCHD = Schizoid Scale, TPQ = Tridimensional Personality Questionnaire, TCI = Temperament and Character Inventory, Beck's Depression Inventory 21 = BDI-21, GAD-7 = Generalized Anxiety Disorder 7-questionnaire, FABQ = Fear-Avoidance Beliefs Questionnaire, PAL = Paired Associates Learning.

(questions on smoking, alcohol, and other intoxicant use), health, economy, work, and mental resources. Depressive and anxiety symptoms were questioned using HSCL-25, the Generalized Anxiety Disorder 7-questionnaire (GAD-7) [25], Beck's Depression Inventory 21 (BDI-21), and psychological features using the Fear-Avoidance Beliefs Questionnaire (FABQ) (Table 1).

The NFBC1966 also contains many subsamples, of which some are focused on psychiatric outcomes. In 1997–1998, a study was conducted on people living in the city of Oulu. The sample was based on the HSCL-25 screening, which was part of a 31-year follow-up. Based on the screening, 234 screen positives (HSCL-25 scores above the mean) and every tenth screen negatives were invited to the Structured Clinical Interview for DSM-III (SCID) [27]. The SCID interview consists of two parts: SCID-I for making major DSM-III-R axis-I diagnoses and SCID-II for diagnosing all DSM-III-R personality disorders. The interview was conducted with 209 screen positives and 112 screen negatives [21]. In 1990–2001 a sub-study of individuals with psychosis and their controls was conducted. Out of the 142 invited individuals with psychosis, 91took part in the sub-study. Comparison subjects were sex-matched randomly selected from NFBC1966 members who were known not to have had a psychotic episode. A total of 187 controls were invited, and 104 subjects participated in the study [28]. The data were collected by SCID-I interview, questionnaires (including questions on use of antipsychotic medication, social background, and substance use), brain magnetic resonance imaging (MRI) scans, and cognitive tests [29]. The follow-up for the above subsample was conducted in 2008–2011, also including new subjects, 107 cases with psychotic disorders, including 54 subjects with schizophrenia, and 194 controls. The follow-up included psychiatric interviews, questionnaires, brain MRI scans, and cognitive tests [30]. In the studies conducted to individuals with psychosis and their controls, 54 cases with psychotic disorder and 76 controls participated in both sub studies. The psychiatric sub-studies and other sub-studies are presented in Table 2.

**Table 2. Sub-studies of the Northern Finland Birth Cohort 1966 (NFBC1966).**

| Age (main reference) | Target Population(N) | Questionnaire data (Participation rate) | Clinical Examination (Participation rate) |
|---|---|---|---|
| 1 year [18] | Children with perinatal risk $n = 793$ | N/A | Neurological examination $n = 722$ (91%) |
| 14 years [31] | Children with low school performance $n = 495$ | N/A | IQ-test |
| 16 years [32] | Unwanted children ($n = 231$) with control group ($n = 227$) | Questionnaire for teachers of unwanted children $n = 88$ (38.1%) and controls $n = 89$ (39.2%) | N/A |
| 15 years [33] | Children with known myopia $n = 707$ and their controls $n = 784$ | N/A | Ophthalmological examination of cases $n = 236$ (33.4%) and controls $n = 266$ (33.9%) |
| 19 years [34] | Twins and their controls $n = 652$ | Questionnaire at age 19 $n = 652$ | N/A |
| 24 years [35] | Random subsample of males $n = 2500$ | On health and life satisfaction $n = 2500$ | N/A |
| 31 years [21] | Living in the city of Oulu $n = 1609$ | SCID II | SCID I and SCID II interviews |
| 31 years [36] | Participated in 31-year clinical examination in the city of Oulu $n = 1609$ | N/A | Bone mineral density and content measurements $n = 1102$ (68.5%) |
| 31 years [36] | Random sample based on 31-years questionnaire $n = 196$ | Exercise and food diaries for 7 days | N/A |
| 31 years[a] | Subjects with problems in hearing at age of 14 and their controls $n = 1372$ | N/A | Audiogram studies |
| 33 years [29] | Subjects with psychosis and their controls $n = 191$ | Psychiatric interview and questionnaire | Brain MRI scans and cognitive test |
| 39 years[a] | All men alive | Postal questionnaire regarding loss of hair ($n = 3128$) | N/A |
| 42 years [29] | Subjects with psychosis, their controls and siblings $n = 312$ | Psychiatric interview and questionnaire | Brain magnetic resonance imaging (MRI) scans and cognitive test |
| 48 years [37] | Random sample of the cohort was invited to eye screening $n = 5155$ | N/A | Eye screenings, $n = 3070$ |

[a]Data has not been published.

## Study and comparison cohort

The study cohort comprises all the people born in the provinces of Oulu and Lapland in 1966. In 1966 there were 12 043 births in the provinces of Lapland and Oulu, of which 96.3% are members of the NFBC1966. The comparison cohort comprises all the people born in the provinces of Lapland and Oulu in 1965 and 1967. The cohort included 24 471 subjects, 12 465 (50.9%) born in 1965 and 12 006 (49.1%) born in 1967. Personal identification numbers and date of death were obtained from the Digital and Population Data Services Agency. Medical history was obtained from the Finnish Institute of Health and Welfare (THL).

From both datasets we included in our analysis those, who were living at Finland in the beginning of follow-up at age 10. In the study cohort n = 53 (0.4%) died before their 10th birthday and n = 543 (4.5%) were living abroad at the age of 10. Respective numbers in the comparison cohort are n = 86 (0.4%) and n = 1046 (4.3%). After exclusions, 11 447 study subjects in the study cohort and 23 339 subjects in the comparison cohort were included in the analysis.

## Use of psychiatric care services

We examined the use of psychiatric care services between 10th and 50th birthday using several variables. First, we created a variable indicating if the study subject had had any visit to psychiatric specialty (outpatient or inpatient). The second variable indicates if the study subject had any psychiatric hospitalization. The third variable indicates the number of days of psychiatric hospitalization of those, who had had any psychiatric hospitalization.

The variables on the use of psychiatric care services were obtained from the Care Register for Health Care (CRHC), maintained by THL. The CRHC is one of the oldest individual-level hospital discharge registers and has nationwide hospital discharge information on inpatient visits starting from 1967 (from 1969 with complete identification numbers). From 1998 onwards, the register also includes information on specialized outpatient care. By law, all hospitals are obligated to report all inpatient care. The outpatient care data covers public hospitals only.

## Statistical analysis

We used several analysis methods to estimate the use of psychiatric care services between the study cohort and comparison cohort. First, we analysed the difference in any visit to any psychiatric specialty and any psychiatric hospitalization between the study and comparison cohorts using the Chi-square test. The number of days of psychiatric hospitalisation was analysed using the Mann-Whitney U-test. Age of first visit to psychiatric specialty and age of first psychiatric hospitalization was analysed using the Welch's t-test.

Then, for the first visit to psychiatric speciality and for the first psychiatric hospitalisation, we fitted the Cox proportional hazards regression models to calculate the hazard ratios (HR) with 95% confidence intervals (CI). We also fitted the Kaplan-Meier curves and calculated the equality of survival -functions using the Log-rank-test. Time of emigration and death were used as a censoring point in analyses (information from the Population Register Centre). The number of days of psychiatric hospitalisation was analysed using the Zero-Truncated Negative Binomial Regression and incidence rate ratios (IRR) with 95% CIs are reported. Data were analysed using R software version 1.1.453.

## Results

Table 3 presents the use of psychiatric care in the study and comparison cohort by sex.

There was no difference between the study and comparison cohort in any visit to psychiatric speciality or in any psychiatric hospitalisation. Among those who had had psychiatric hospitalisation there were statistically significant results for days in psychiatric hospitalisation: males in the study cohort had a lower median number of days of psychiatric hospitalisation (study cohort median 28 days, comparison cohort median 31 days ($U = 1.51 \times 10^5$, p < .001)), whereas females in the study cohort had a higher median number of days of psychiatric hospitalisation than the comparison cohort (study cohort median 39, comparison cohort median 38

**Table 3. Psychiatric care service use in study and comparison cohort in males and females.**

| | Male | | | | Female | | | |
|---|---|---|---|---|---|---|---|---|
| | *n* = 17 744 | | | | *n* = 17 042 | | | |
| | **Study** | **Comparison** | | | **Study** | **Comparison** | | |
| | **n = 5838** | **n = 11 906** | | | **n = 5609** | **n = 11 433** | | |
| | N (%) | N (%) | $\chi^2$ | *p* | N (%) | N (%) | $\chi^2$ | *p* |
| Visit to psychiatric speciality | 744 (12.7) | 1434 (12.0) | 1.72 | 0.190 | 727 (13.0) | 1510 (13.2) | 0.18 | 0.627 |
| Age of first visit to psychiatric specialty[a] | 35.14 (9.70) | 35.07 (9.24) | 0.15 | 0.875 | 36.90 (8.53) | 37.50 (8.55) | -1.58 | 0.116 |
| Psychiatric hospitalisation | 457 (7.8) | 867 (7.3) | 2.49 | 0.115 | 312 (5.6) | 654 (5.6) | 0.13 | 0.716 |
| Age of first psychiatric hospitalisation[a] | 31.20 (9.67) | 31.14 (9.20) | 0.10 | 0.918 | 32.42 (9.97) | 32.33 (9.68) | 0.13 | 0.895 |
| Days of psychiatric hospitalisation[b] | 28 (7.0–89.5) | 31 (8–89) | $1.51 \times 10^5$ | < .001 | 39 (11–131) | 38 (11–141.9) | $7.57 \times 10^4$ | < .001 |

[a]Mean (SD), Analysed with Welch's t-test.
[b]Median (IQR), Analysed with Mann-Whitney U-statistics.

**Table 4. Results of comparison between study and comparison cohorts analysed with Cox Proportional Hazard model (HR) and Zero-Truncated Negative Binomial regression (IRR) with 95% CI. by sex.** The comparison cohort is the reference group in all models.

| | Male | | | Female | | |
| --- | --- | --- | --- | --- | --- | --- |
| | (*n* = 17 744) | | | (*n* = 17 042) | | |
| | **HR** | **95% CI** | *p* | **HR** | **95% CI** | *p* |
| First visit to psychiatric specialty | | | | | | |
| Study cohort | 1.05 | 0.96–1.15 | 0.244 | 0.98 | 0.90–1.07 | 0.672 |
| Hospitalization in a psychiatric specialty | | | | | | |
| Study cohort | 1.09 | 0.97–1.22 | 0.133 | 1.03 | 0.90–1.17 | 0.705 |
| | **IRR** | **95% CI** | *p* | **IRR** | **95% CI** | *p* |
| Days of psychiatric hospitalization | | | | | | |
| Study cohort | 0.92 | 0.74–1.17 | 0.530 | 0.86 | 0.68–1.10 | 0.231 |

days ($U = 7.57{\times}10^4$, p $<$ .001)). Age of first visit to psychiatric specialty or age of first psychiatric hospitalization did not differ between the study and comparison cohorts (Table 3).

Table 4 presents the results of the Zero Truncated Negative Binomial Regression and Cox Proportional Hazard regression. No difference was found in the use of psychiatric care services between the study and comparison.

The results of the Kaplan-Meier survival curves are presented in Figs 1 and 2.

The survival curves for first visit to psychiatric speciality (Fig 1) or first psychiatric hospitalisation (Fig 2) between the study and comparison cohorts did not differ.

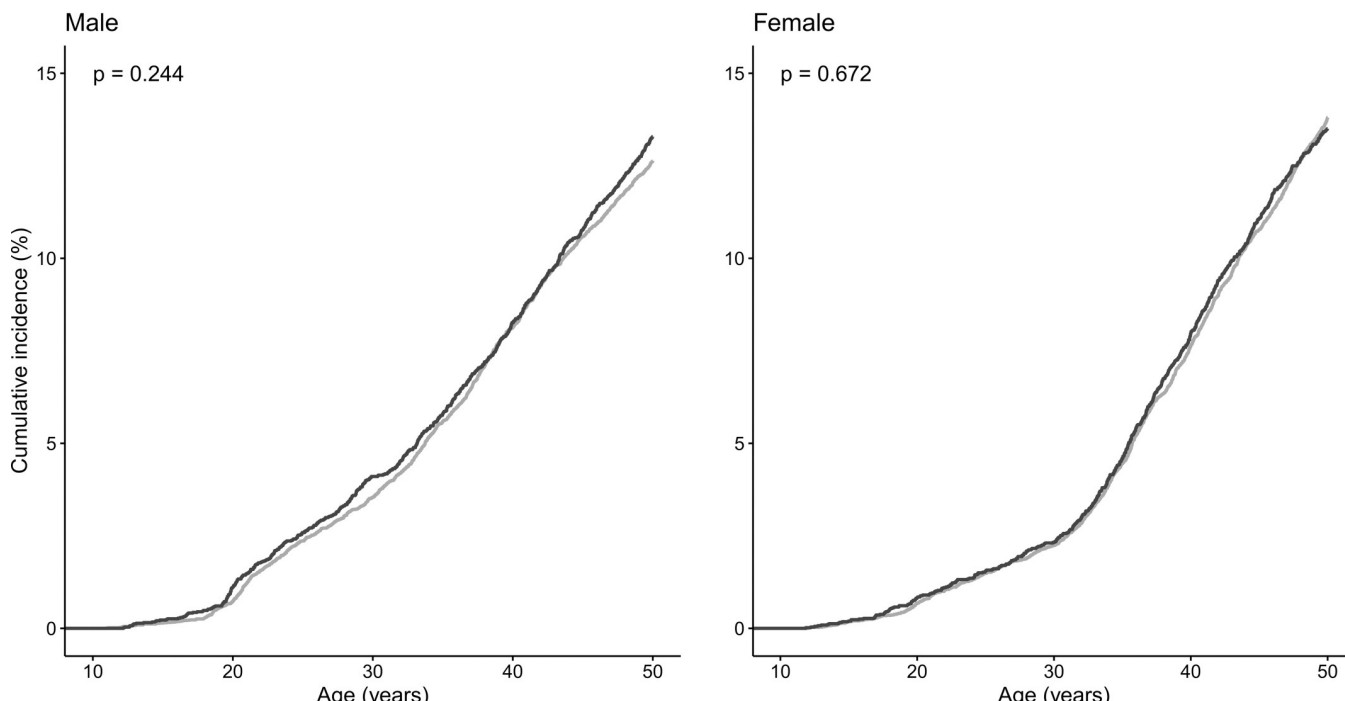

**Fig 1. Kaplan-Meier curve for first visit to psychiatric specialty for study and comparison cohorts by sex with p-value from log-rank test.**

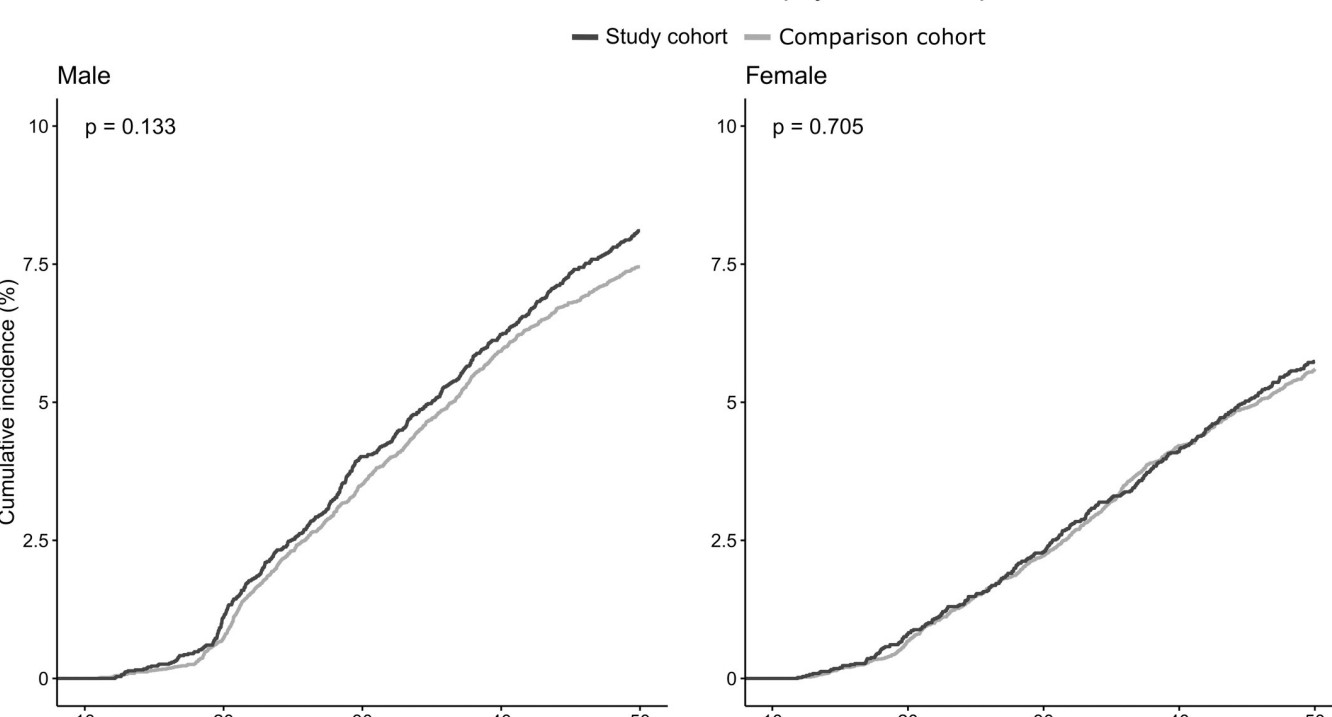

**Fig 2. Kaplan-Meier curve for first psychiatric hospitalization for study and comparison cohorts by sex with p-value from log-rank test.**

## Discussion

No evidence of an association of being a member of a longitudinal follow-up study and studies conducted during follow-up to a use of psychiatric care services was found. There was a significant result on the median number of days of psychiatric hospitalisation, but when analysed further with the Zero-Truncated Binomial Regression, the significance diminished. The results indicate that even a long follow-up with many follow-ups does not necessarily affect participants' life or behaviour, at least in the use of psychiatric care service. Of those 12.7% who visited psychiatric specialty, the mean ages of first visit for psychiatric specialty were between 35 and 37 years and of those 6.7% who had psychiatric hospitalization, mean ages of first psychiatric hospitalizations were between 31 and 32 years in the study cohort. The follow-ups for whole cohort including psychiatric screens were conducted at age 31 and 46, and the sub-studies focusing on psychiatric measurements at age 33 and 42. The associations of participation in an epidemiological follow-up study are rarely studied, and the results need to be replicated. The NFBC1966 may be regarded as representative at the population level in terms of psychiatric outcomes despite the personal follow-up of the birth cohort.

The participation rate in the follow-up studies decreased from 91.2% in the 1-year study, 93.6% in the 14-year study, 77.4% in the 31-year study to 66.5% in the 46-year study. It is already known that participants of the NFBC1966 in the 31-year and 46-year studies are more often female, were less often unemployed and are from higher social class [26] and subjects with psychiatric disorders participated less actively than those without any psychiatric disorders in the 31-year study [38]. The selective non-participation bias does exist in the NFBC1966. However, as the aim of the study was to examine the effects of participation in the

longitudinal follow-up study, and the postal follow-up questionnaires have been sent to
N = 11,870 (97.0% of original cohort members) at 1-year follow-up, N = 11,768 (96.2%) at
14-year follow-up, N = 11,543 (94.4%) at 31-year follow-up and N = 10,331 (83.8%) at age of
46-years, the vast majority of members of NFBC1966 have been aware of longitudinal follow-
up.

A previous study did find an association between the Northern Finland Birth Cohort 1986
(NFBC1986) and mental disorders; the female members of the NFBC1986 had less psychiatric
diagnoses at age 2 to 28 years than the comparison cohort [39]. Both of the NFBC studies are
prospective longitudinal birth cohorts with multiple follow-ups. Even though both cohorts
have been followed since birth, in the NFBC1986 the follow-up has been more intense in child-
hood and adolescence (follow-ups for the whole cohort at age 7, 8 and 15–16), while in the
NFBC1966 there was data collection only at age 14 with a questionnaire. The difference in fol-
low-ups might explain the contradictive results in the studies.

The UKKI-study, in which the study participants had used mental health care services
more often than the control group, was a long follow-up psychiatric study aiming to find peo-
ple with psychiatric disorders using psychiatric interviews [8]. Compared to the UKKI study
the NFBC1966 did not focus on psychiatric outcomes but covered all aspects of health-related
issues. In the NFBC1966 only a minority of the participants were interviewed using psychiatric
methods. In the psychiatric UKKI-study, participants were interviewed twice, which might
have reduced the unmet need for psychiatric treatment by motivating people to seek treatment
for their psychiatric ill-health [5,6].

### Strengths

To our knowledge, the associations of participation in a prospective epidemiological study are
rarely studied. The size of the study population was relatively large, and the follow-up covered
40 years. The data used in this study were from the Finnish Care Register for Health Care,
which has been found to have a good standard [40].

The participation rate in the NFBC1966 can be considered somewhat high in the oldest fol-
low-ups (93.6% in the 14-year follow-up and 77.4% in the 31-year follow-up), which makes the
study setting even stronger. Nowadays there is an issue of poor response rates in follow-up
population cohort studies [41].

### Limitations

The study also has some limitations. We could not identify those in our study cohort who
really participated in the NFBC1966 and took part to follow-up studies. Nevertheless, 96.3% of
our study cohort was originally a member of the NFBC1966. Even though the follow-up in
NFBC includes many questionnaires and clinical examinations, only few of them were focused
on psychiatric measurements. Also, we did not have the information of possible family mem-
bers in the study and comparison cohorts. The health care use of a family member may affect
other family members.

### Conclusions

We found no evidence of an association between participation in an epidemiological follow-
up study and the use of psychiatric care services. The Northern Finland Birth Cohort 1966
may be regarded as representative at the population level in terms of psychiatric outcomes
despite the personal follow-up of the birth cohort. The effects of participation in epidemiologi-
cal follow-up studies have previously been under-examined, and the results need to be repli-
cated. In addition, the use of psychiatric medications might prove an important area for future

research. The follow-up studies conducted among the participants might be considered as interventions.

## Acknowledgments

We thank all cohort members and researchers who participated in the Northern Finland Birth cohort 1966 study. We also wish to acknowledge the work of the NFBC project center.

## Author Contributions

**Conceptualization:** Martta Kerkelä, Mika Gissler, Juha Veijola.

**Data curation:** Martta Kerkelä, Tanja Nordström.

**Formal analysis:** Martta Kerkelä.

**Methodology:** Martta Kerkelä.

**Project administration:** Juha Veijola.

**Resources:** Juha Veijola.

**Supervision:** Mika Gissler, Juha Veijola.

**Visualization:** Martta Kerkelä.

**Writing – original draft:** Martta Kerkelä.

**Writing – review & editing:** Martta Kerkelä, Mika Gissler, Tanja Nordström, Juha Veijola.

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
