## [Decision Letter · Decision Letter 0]

2 Sep 2022

PONE-D-22-16407Association between participation to the Northern Finland Birth Cohort 1966 study and use of psychiatric care servicesPLOS ONE

Dear Dr. Kerkelae,

Thank you for submitting your manuscript to PLOS ONE. After careful consideration, we feel that it has merit but does not fully meet PLOS ONE’s publication criteria as it currently stands. Therefore, we invite you to submit a revised version of the manuscript that addresses the points raised during the review process.

 Two external reviewers have evaluated your submission. Both were positive about the manuscript, but Reviewer 1 has made some suggestions for improvement that we would ask you to consider and respond to. Please attend carefully to each of the points they have raised when preparing your revision.

We look forward to receiving your revised manuscript.

Kind regards,

Jamie Males

Editorial Office

PLOS ONE

Journal Requirements:

Reviewers' comments:

Reviewer's Responses to Questions

**Comments to the Author**

1. Is the manuscript technically sound, and do the data support the conclusions?

Reviewer #1: Yes

Reviewer #2: Yes

2. Has the statistical analysis been performed appropriately and rigorously? 

Reviewer #1: Yes

Reviewer #2: Yes

3. Have the authors made all data underlying the findings in their manuscript fully available?

Reviewer #1: No

Reviewer #2: No

4. Is the manuscript presented in an intelligible fashion and written in standard English?

Reviewer #1: Yes

Reviewer #2: No

5. Review Comments to the Author

Reviewer #1: The study by Dr. Kerkelae and colleagues aimed to examine whether the use of psychiatric care services differed between individuals included in the Northern Finland Birth Cohort 1966 (NFBC1966) and individuals born in 1965 and 1967. The authors found no evidence of a difference in psychiatric outcomes, suggesting that the NFBC1966 may represent the population. The results may be relevant for future research using the NFBC1966 cohort. I have a few suggestions/comments the authors may wish to consider:

1. Table 3: Suggest presenting the results in the form of N (%)

2. How many comparison individuals born in 1965 and 1967 are family members (siblings or a spouse) of individuals in the NFBC1966 cohort? The visit of a family member may affect their health service use and thus dilute the impact of the cohort effect.

3. As the authors discussed in the limitation section, only a few follow-ups focused on psychiatric measurements at ages 33 and 42 years. What was the mean (SD) age of first psychiatric hospital contact? Suppose the first visit to a psychiatric specialty occurred before these follow-ups. In that case, it is less likely affected by the follow-up.

4. The authors studied severe psychiatric outcomes, such as psychiatric specialty visits and hospitalizations. However, it is relevant to know whether the patterns of prescription filling for psychiatric medications differ between individuals in the NFBC1966 and individuals born in 1965 or 1967.

Reviewer #2: Thank you for letting me review this article. I suggest accept. I believe this submission will contribute to fill the gap in knowledge in whether how much questionnaires in research intervene in the health care seeking patterns.

6. PLOS authors have the option to publish the peer review history of their article (what does this mean?). If published, this will include your full peer review and any attached files.

Reviewer #1: No

Reviewer #2: No

---

## [Author Response · Author response to Decision Letter 0]

26 Sep 2022

Comments by the Academic Editor:

Editor E1. Please ensure that your manuscript meets PLOS ONE's style requirements, including those for file naming. The PLOS ONE style templates can be found at

Response E1. We have now revised our manuscript and restyled the manuscript and files. 

Editor E2. In your Data Availability statement, you have not specified where the minimal data set underlying the results described in your manuscript can be found. PLOS defines a study's minimal data set as the underlying data used to reach the conclusions drawn in the manuscript and any additional data required to replicate the reported study findings in their entirety. All PLOS journals require that the minimal data set be made fully available. For more information about our data policy, please see http://journals.plos.org/plosone/s/data-availability.

Response E2: We have now modified the Data Availability Statement:

‘Data cannot be shared publicly because the utilised data have been given for this specific study (Permission numbers: TK-53-1053-19, VRK/3285/2019-2, THL/1254/2019), and the data cannot be shared without permission from the register keepers. The same data can be applied from Findata, the Finnish Health and Social Data Permit Authority by researchers who meet the criteria for access to confidential data. Additionally, Findata, can prepare statistical tables from the registers used in this study: 

https://findata.fi/en/permits/data-requests/’

Editor E3. Please review your reference list to ensure that it is complete and correct. If you have cited papers that have been retracted, please include the rationale for doing so in the manuscript text, or remove these references and replace them with relevant current references. Any changes to the reference list should be mentioned in the rebuttal letter that accompanies your revised manuscript. If you need to cite a retracted article, indicate the article’s retracted status in the References list and also include a citation and full reference for the retraction notice.

Response E3: We revised the reference list (we did not identify any retracted references) and made corrections to the references: ref 7 and ref 28 (added the doi), ref 18, ref 20, ref 30 (corrected the doi) and corrected the title of the reference 32. 

 

Comments by the Rewievers 

Reviewer #1

Reviewer R1.1: The study by Dr. Kerkelä and colleagues aimed to examine whether the use of psychiatric care services differed between individuals included in the Northern Finland Birth Cohort 1966 (NFBC1966) and individuals born in 1965 and 1967. The authors found no evidence of a difference in psychiatric outcomes, suggesting that the NFBC1966 may represent the population. The results may be relevant for future research using the NFBC1966 cohort.

Response R1.1: Thank you for this comment. 

Reviewer R1.1: Table 3: Suggest presenting the results in the form of N (%)

Response R1.1: We have now modified the table and report the results in the form of N (%).

Reviewer R1.2. How many comparison individuals born in 1965 and 1967 are family members (siblings or a spouse) of individuals in the NFBC1966 cohort? The visit of a family member may affect their health service use and thus dilute the impact of the cohort effect.

Response R1.2: Unfortunately, we do not have data of family members. We agree that this is a potential limitation of the study, and added the information in the limitations section: 

‘Also, we did not have the information of possible family members in the study and comparison cohorts. The health care use of a family member may affect other family members.’. 

Reviewer R1.3. As the authors discussed in the limitation section, only a few follow-ups focused on psychiatric measurements at ages 33 and 42 years. What was the mean (SD) age of first psychiatric hospital contact? Suppose the first visit to a psychiatric specialty occurred before these follow-ups. In that case, it is less likely affected by the follow-up.

Response R1.3: We added the mean ages with SDs and the comparison with the Welch's t-test of the age of first visit to psychiatric specialty and first psychiatric hospitalisation to the table 3 and edited the legend of the table: 

‘Psychiatric care service use in study and comparison cohort in males and females.’ 

The new analysis is mentioned in the 2.4 Statistical analysis section:

‘Age of first visit to psychiatric specialty and age of first psychiatric hospitalization was analysed using the Welch's t-test’ 

and in the 3. Results section the new comparison was mentioned:

‘Age of first visit to psychiatric specialty or age of first psychiatric hospitalization did not differ between the study and comparison cohorts.’ 

In the 4. discussion section the mean ages are mentioned: 

‘Of those 12.7% who visited psychiatric specialty, the mean ages of first visit for psychiatric specialty were between 35 and 37 years and of those 6.7% who had psychiatric hospitalization, mean ages of first psychiatric hospitalizations were between 31 and 32 years in the study cohort. The follow-ups for whole cohort including psychiatric screens were conducted at age 31 and 46, and the sub-studies focusing on psychiatric measurements at age 33 and 42. ´

Reviewer R1.4. The authors studied severe psychiatric outcomes, such as psychiatric specialty visits and hospitalizations. However, it is relevant to know whether the patterns of prescription filling for psychiatric medications differ between individuals in the NFBC1966 and individuals born in 1965 or 1967.

Response R1.4: We agree with the reviewer that further elaborating on this point would be important. Unfortunately, we do not have data of the medications, since the registration in Finland was started in 1994 (reimbursed medication) and it was not complete before 2017 (prescribed medication). We have added the potential future research in the conclusion: 

‘The effects of participation in epidemiological follow-up studies have previously been under-examined earlier, and the results need to be replicated. In addition, the use of psychiatric medications might prove an important area for future research.’ 

Reviewer R2: Thank you for letting me review this article. I suggest accept. I believe this submission will contribute to fill the gap in knowledge in whether how much questionnaires in research intervene in the health care seeking patterns.

Response R2: We appreciate the Reviewer’s input to review the manuscript and give this positive comment. The manuscript has now been proofread, and the typographical and grammatical errors have been corrected as you suggested. 

Additional edits:

 A1. The manuscript has been proofread by a professional.

---

## [Decision Letter · Decision Letter 1]

28 Nov 2022

PONE-D-22-16407R1Association between participation in the Northern Finland Birth Cohort 1966 study and use of psychiatric care servicesPLOS ONE

Dear Dr. Kerkelae,

Thank you for submitting your manuscript to PLOS ONE. After careful consideration, we feel that it has merit but does not fully meet PLOS ONE’s publication criteria as it currently stands. Therefore, we invite you to submit a revised version of the manuscript that addresses the points raised during the review process.

In merit  to your significant work and respecting the concerns raised by the third reviewer,I would like to invite your attention on  a few points:1. Please clarify if there might be any bias or interpretation between the association of participation of psychiatric patients and non-participants in the study.2. Please give an an explanation or write a paragraph that could explain if there emerge any difference when the comparison cohort study is conducted  with a   reduced sample. Please pay attention to the reviewers' comments and suggestions, and re-submit a revised until Jan 12 2023 11:59PM, so we will manage to review it with the proposed changes at the soonest.  If you will need more time than this to complete your revisions, please reply to this message or contact the journal office at plosone@plos.org. Please include the following items when submitting your revised manuscript:A rebuttal letter that responds to each point raised by the academic editor and reviewer(s). You should upload this letter as a separate file labeled 'Response to Reviewers'.A marked-up copy of your manuscript that highlights changes made to the original version. You should upload this as a separate file labeled 'Revised Manuscript with Track Changes'.An unmarked version of your revised paper without tracked changes. You should upload this as a separate file labeled 'Manuscript'.

We look forward to receiving your revised manuscript.

Kind regards,

Dr. Silva Ibrahimi, PhD

Academic Editor

PLOS ONE

Journal Requirements:

Reviewers' comments:

Reviewer's Responses to Questions

**Comments to the Author**

1. If the authors have adequately addressed your comments raised in a previous round of review and you feel that this manuscript is now acceptable for publication, you may indicate that here to bypass the “Comments to the Author” section, enter your conflict of interest statement in the “Confidential to Editor” section, and submit your "Accept" recommendation.

Reviewer #2: All comments have been addressed

Reviewer #3: (No Response)

2. Is the manuscript technically sound, and do the data support the conclusions?

The manuscript must describe a technically sound piece of scientific research  data that supports the conclusions. Experiments must have been conducted rigorously, with appropriate controls, replication, and sample sizes. The conclusions must be drawn appropriately based on the data presented. 

Reviewer #2: Yes

Reviewer #3: Yes

3. Has the statistical analysis been performed appropriately and rigorously? 

Reviewer #2: Yes

Reviewer #3: Yes

4. Have the authors made all data underlying the findings in their manuscript fully available?

Reviewer #2: No

Reviewer #3: No

5. Is the manuscript presented in an intelligible fashion and written in standard English?

Reviewer #2: Yes

Reviewer #3: Yes

6. Review Comments to the Author

Reviewer #2: (No Response)

Reviewer #3: In this study, the authors examined whether use of psychiatric care services differed between individuals included in the Northern Finland Birth Cohort 1966 (NFBC1966) differed from individuals born in 1965 and 1967, their hypothesis being that study involvement may act as an intervention of sorts by motivating people to seek treatment they otherwise would not. The authors found no evidence of a difference in psychiatric outcomes, suggesting NFBC1966 is representative of the broader population. I think this could be valuable information for those using this data resource in the future. However, I have a couple of things I would like to see the authors address first...

1. The authors' characterization of potential inks between participation and health implies that participation may cause improvement in health. This seems a little odd to me, typically associations between participation and health are interpreted through the lens of selection bias - unhealthy people are less likely to take part in studies and more likely to drop out when they do. This doesn't actually impact the value of the research, but the authors should acknowledge that where associations are detected elsewhere this is often the interpretation.

2. Analyses all compare the full 1966 cohort to the combined 1965/67 cohort. However, the 1966 cohort actually diminishes over time, as reduced numbers take part at age 1, 14, 31, and 46. Can the authors compare the reduced samples to the comparison cohort to see if differences emerge over time? This again speaks to the fact that differences between participants and non-participants are typically interpreted as selection bias, rather than evidence for positive effects of the study's participation. e.g. The authors hypothesize that study participation will increase use of psychiatric services, so the 1966 cohort will use more that the 65/67 cohort. They find no evidence for this. Would their hypothesis apply to the samples reduced by attrition? i.e. if participation increases service use then would continued participation predict service use? This would run counter to a hypothesis that selection bias would mean that service use would predict drop out. Could the authors run such analyses to test these contrasting hypotheses?

7. PLOS authors have the option to publish the peer review history of their article (what does this mean?). If published, this will include your full peer review and any attached files.

Reviewer #2: No

Reviewer #3: No

---

## [Author Response · Author response to Decision Letter 1]

20 Dec 2022

Comments by the Academic Editor:

Editor E1: Please clarify if there might be any bias or interpretation between the association of participation of psychiatric patients and non-participants in the study.

Response E1: We are aware of selective non-participation bias in the NFBC1966. It is already known that participants of the NFBC1966 in the 31-year and 46-year studies are more often female, were less often unemployed and are from higher social class (Nordström T, Miettunen J, Auvinen J et al., 2021) and subjects with psychiatric disorders participated less actively than those without any psychiatric disorders in the 31-year study (Haapea M, Miettunen J, Läärä E, et al. 2008). But, in the present study we did not actually focus on the attrition of the NFBC1966 sample, but more in the effects of the longitudinal follow-up study, nevertheless have the subject participated in each follow-up study or not. Please see response R3.3. 

References:

 Nordström T, Miettunen J, Auvinen J, Ala-Mursula L, Keinänen-Kiukaanniemi S, Veijola J, et al. Cohort Profile: 46 years of follow-up of the Northern Finland Birth Cohort 1966 (NFBC1966). Int J Epidemiol. 2021; dyab109. doi:10.1093/ije/dyab109

Haapea M, Miettunen J, Isohanni MK, Veijola JM, Läärä E, Järvelin MR, et al. Non-participation in a field survey with respect to psychiatric disorders. Scand J Public Health. 2008;36. doi:10.1177/1403494808092250

Editor E2: Please give an an explanation or write a paragraph that could explain if there emerge any difference when the comparison cohort study is conducted with a reduced sample.

Response E2: The study group was not NFBC1966 per se, but a population born in 1966 in Northern Finland. We could not identify those who participated in the NFBC1966 and took part in follow-up studies. Nevertheless, 96.3% of the study cohort were members of the NFBC1966. Please see response R3.3

Journal Requirements JR1: Please review your reference list to ensure that it is complete and correct. If you have cited papers that have been retracted, please include the rationale for doing so in the manuscript text, or remove these references and replace them with relevant current references. Any changes to the reference list should be mentioned in the rebuttal letter that accompanies your revised manuscript. If you need to cite a retracted article, indicate the article’s retracted status in the References list and also include a citation and full reference for the retraction notice.

Response JR1: We have now revised the reference list and corrected the reference 20: Järvelin MR, Sovio U, King V, Lauren L, Xu B, McCarthy MI, Hartikainen AL, Laitinen J, Zitting P, Rantakallio P, Elliott P. Early Life Factors and Blood Pressure at Age 31 Years in the 1966 Northern Finland Birth Cohort. Hypertension. 2004;44: 838–846. doi:10.1161/01.HYP.0000148304.33869.ee

Reviewer R3.1: In this study, the authors examined whether use of psychiatric care services differed between individuals included in the Northern Finland Birth Cohort 1966 (NFBC1966) differed from individuals born in 1965 and 1967, their hypothesis being that study involvement may act as an intervention of sorts by motivating people to seek treatment they otherwise would not. The authors found no evidence of a difference in psychiatric outcomes, suggesting NFBC1966 is representative of the broader population. I think this could be valuable information for those using this data resource in the future. However, I have a couple of things I would like to see the authors address first...

Response R3.1: Thank you for the comment. 

Reviewer R3.2: The authors' characterization of potential inks between participation and health implies that participation may cause improvement in health. This seems a little odd to me, typically associations between participation and health are interpreted through the lens of selection bias - unhealthy people are less likely to take part in studies and more likely to drop out when they do. This doesn't actually impact the value of the research, but the authors should acknowledge that where associations are detected elsewhere this is often the interpretation.

Response R3.2 The reviewer is right, the selection bias exists in the NFBC1966 (Please see a more detailed answer for attrition and biases in Response R3.3). We were focused on the effects of being a member of the longitudinal follow-up study, more than attrition. We modified the aims of the study: 

‘We examined the effects of being a member of the longitudinal follow-up study cohort compared to a population born in the same geographical area in Northern Finland in 1965 and 1967.’ 

edited the abstract: ‘Although the idea is not to intervene, being a member of the longitudinal follow-up study and studies conducted during follow-up may affect the target population. ‘ 

introduction: ‘Although the idea is not to intervene, being a member of the longitudinal follow-up study and studies conducted during follow-up may affect the study population in various ways.’

also we added achapter on the effects of awareness of participation in the study in the introduction section to emphasize the aims of the study: 

‘The awareness of participation in the study can be described as the Hawthorne effect, which describes the awareness of the study participants being studied and the possible impact on their beaviour. (Jones, 1992) In the Derbyshire Smoking Study, the Hawthorne effect of repeatedly measured smoking behaviour of about 6000 adolescents was studied by questionnaire yearly from 1974 to 1978 in selected schools. The self-reported smoking habits were compared to adolescents whose smoking habits were measured first time in 1978. The findings revealed that the prevalence of smoking was lower in those schools, which had been surveyed for five years. (Murray et al., 1988). Berkhout et all. pooled the findings of 15 studies published in 2012-2022 on the Hawthorne effect in primary care or outpatient clinics or healthy subjects. In the meta-analysis, the Hawthorne effect was defined as “an aware or unconscious complex behavioural change in a study environment, related to the interaction of four biases affecting the study subjects and investigators: selection bias, commitment and congruence bias, conformity and social desirability bias, and observation and measurement bias”. The findings demonstrate that medical research is unavoidably prone to the Hawthorne effect, which restricts its external validity, beginning with the intentional or unintentional selection of the study population, resulting in blind spots in medical knowledge. (Berkhout et al., 2022)’

Modified the discussion:

‘No evidence of an association of being a member of a longitudinal follow-up study and studies conducted during follow-up to a use of psychiatric care services was found.’

Reviewer R3.3. Analyses all compare the full 1966 cohort to the combined 1965/67 cohort. However, the 1966 cohort actually diminishes over time, as reduced numbers take part at age 1, 14, 31, and 46. Can the authors compare the reduced samples to the comparison cohort to see if differences emerge over time? This again speaks to the fact that differences between participants and non-participants are typically interpreted as selection bias, rather than evidence for positive effects of the study's participation. e.g. The authors hypothesize that study participation will increase use of psychiatric services, so the 1966 cohort will use more that the 65/67 cohort. They find no evidence for this. Would their hypothesis apply to the samples reduced by attrition? i.e. if participation increases service use then would continued participation predict service use? This would run counter to a hypothesis that selection bias would mean that service use would predict drop out. Could the authors run such analyses to test these contrasting hypotheses?

Response R3.3. Our study group was not NFBC1966 per se, but a population born in 1966 in Northern Finland. We could not identify those in our study cohort who really participated in the NFBC1966 and took part in follow-up studies. Nevertheless, 96.3% of our study cohort were members of the NFBC1966. 

We are aware of selective non-participation bias in the NFBC1966. It is already known that participants of the NFBC1966 in the 31-year and 46-year studies are more often female, were less often unemployed and are from higher social class (Nordström T, Miettunen J, Auvinen J, et al., 2021) and subjects with psychiatric disorders participated less actively than those without any psychiatric disorders in the 31-year study (Haapea M, Miettunen J, Läärä E, et al. 2008) We did not focus in the attrition of the NFBC1966 sample, but more in the effects of being a member in a longitudinal follow-up study, nevertheless have the subject participated in each follow-up study or not. As the postal follow-up questionnaires have been sent to N=11,768 original cohort members at 1-year follow-up, N=11,768 at 14-year follow-up, N=11,543 at 31-year follow-up and N=10,331 at age of 46-years, the vast majority of members of NFBC1966 have been aware of longitudinal follow-up.

We now have discussed the attrition and source of bias in the discussion section: 

‘The participation rate in the follow-up studies decreased from 91.2% in the 1-year study, 93.6% in the 14-year study, 77.4% in the 31-year study to 66.5% in the 46-year study. It is already known that participants of the NFBC1966 in the 31-year and 46-year studies are more often female, were less often unemployed and are from higher social class [26] and subjects with psychiatric disorders participated less actively than those without any psychiatric disorders in the 31-year study [38]. The selective non-participation bias does exist in the NFBC1966. However, as the aim of the study was to examine the effects of participation in the longitudinal follow-up study, and the postal follow-up questionnaires have been sent to N=11,870 (97.0% of original cohort members) at 1-year follow-up, N=11,768 (96.2%) at 14-year follow-up, N=11,543 (94.4%) at 31-year follow-up and N=10,331 (83.8%) at age of 46-years, the vast majority of members of NFBC1966 have been aware of longitudinal follow-up.’

References:

Nordström T, Miettunen J, Auvinen J, Ala-Mursula L, Keinänen-Kiukaanniemi S, Veijola J, et al. Cohort Profile: 46 years of follow-up of the Northern Finland Birth Cohort 1966 (NFBC1966). Int J Epidemiol. 2021; dyab109. doi:10.1093/ije/dyab109

Haapea M, Miettunen J, Isohanni MK, Veijola JM, Läärä E, Järvelin MR, et al. Non-participation in a field survey with respect to psychiatric disorders. Scand J Public Health. 2008;36. doi:10.1177/1403494808092250

---

## [Decision Letter · Decision Letter 2]

22 Feb 2023

Association between participation in the Northern Finland Birth Cohort 1966 study and use of psychiatric care services

PONE-D-22-16407R2

Dear Dr. Kerkelae,

We’re pleased to inform you that your manuscript has been judged scientifically suitable for publication and will be formally accepted for publication once it meets all outstanding technical requirements.Thank you for submitting your work with PLOS and the extraordinary work you introduced.

Please note that I have also acted as Rewier 2 of your submission.

Kind regards,

Silva Ibrahimi, PhD

Academic Editor

PLOS ONE

Additional Editor Comments (optional):

Reviewers' comments:

Reviewer's Responses to Questions

**Comments to the Author**

1. If the authors have adequately addressed your comments raised in a previous round of review and you feel that this manuscript is now acceptable for publication, you may indicate that here to bypass the “Comments to the Author” section, enter your conflict of interest statement in the “Confidential to Editor” section, and submit your "Accept" recommendation.

Reviewer #2: All comments have been addressed

Reviewer #4: All comments have been addressed

2. Is the manuscript technically sound, and do the data support the conclusions?

Reviewer #2: Yes

Reviewer #4: Yes

3. Has the statistical analysis been performed appropriately and rigorously? 

Reviewer #2: Yes

Reviewer #4: Yes

4. Have the authors made all data underlying the findings in their manuscript fully available?

Reviewer #2: No

Reviewer #4: Yes

5. Is the manuscript presented in an intelligible fashion and written in standard English?

Reviewer #2: Yes

Reviewer #4: (No Response)

6. Review Comments to the Author

Reviewer #2: I had already accepted the paper once before

Reviewer #4: The authors addressed all the previous comments raised in a very comprehensive and scientific approach. I congratulate them for the extraordinary work done in the cohort-study work and the technical strength given throughout the paper.

7. PLOS authors have the option to publish the peer review history of their article (what does this mean?). If published, this will include your full peer review and any attached files.

Reviewer #2: No

Reviewer #4: No

---

## [Editor Report · Acceptance letter]

24 Feb 2023

PONE-D-22-16407R2 

Association between participation in the Northern Finland Birth Cohort 1966 study and use of psychiatric care services 

Dear Dr. Kerkelae:

I'm pleased to inform you that your manuscript has been deemed suitable for publication in PLOS ONE. Congratulations! Your manuscript is now with our production department. 

Kind regards, 

on behalf of

Dr. Silva Ibrahimi 

Academic Editor

PLOS ONE